# Monitoring chronic inflammatory musculoskeletal diseases mixing virtual and face-to-face assessments—Results of the digireuma study

**Diego Benavent**[1]*, **Luis Fernández-Luque**[2], **Francisco J. Núñez-Benjumea**[3], **Victoria Navarro-Compán**[1], **María Sanz-Jardón**[1], **Marta Novella-Navarro**[1], **Pedro L. González-Sanz**[2], **Enrique Calvo-Aranda**[4], **Leticia Lojo**[4], **Alejandro Balsa**[1], **Chamaida Plasencia-Rodríguez**[1]

**1** Hospital Universitario La Paz, IdiPaz, Department of Rheumatology, Madrid, Spain, **2** AdheraHealth Inc., Palo Alto, United States of America, **3** Virgen Macarena University Hospital, Innovation Unit, Seville, Spain, **4** Hospital Universitario Infanta Leonor, Department of Rheumatology, Madrid, Spain

* d_benavent@hotmail.com

**Data Availability Statement:** Data are not publicly available as the scarce number of patients, together

## Abstract

Mobile health technology holds great promise for the clinical management of patients with chronic disease. However, evidence on the implementation of projects involving digital health solutions in rheumatology is scarce. We aimed to study the feasibility of a hybrid (virtual and face-to-face) monitoring strategy for personalized care in rheumatoid arthritis (RA) and spondyloarthritis (SpA). This project included the development of a remote monitoring model and its assessment. After a focus group with patients and rheumatologists, relevant concerns regarding the management of RA and SpA were raised, leading to the development of the Mixed Attention Model (MAM), which combined hybrid (virtual and face-to-face) monitoring. Then, a prospective study using the mobile solution Adhera for Rheumatology was conducted. Over a 3-month follow-up period, patients were given the opportunity to complete disease-specific electronic patient reported outcomes (ePROs) for RA and SpA with a pre-established frequency, as well as flares and changes in medication at any time. Number of interactions and alerts were assessed. The usability of the mobile solution was measured by the Net-Promoter Score (NPS) and through a 5-star Likert scale. Following the MAM development, forty-six patients were recruited to utilize the mobile solution, of whom 22 had RA and 24 SpA. There were 4,019 total interactions in the RA group, and 3,160 in the SpA group. Fifteen patients generated a total of 26 alerts, of which 24 were flares and 2 were medication-related problems; most (69%) were managed remotely. Regarding patient satisfaction, 65% of the respondents were considered to have endorsed Adhera for Rheumatology, yielding a NPS of 57 and an overall rating was 4.3 out of 5 stars. We concluded that the use of the digital health solution is feasible in clinical practice to monitor ePROs for RA and SpA. Next steps involve the implementation of this telemonitoring method in a multicentric setting.

with the description of characteristics, make individual patients trackable even after pseudonimization. Signed patient consent forms also did not grant public sharing of patient data. Researchers interested in using the data collected during the study should contact the Ethics Committee of Hospital La Paz (ceic.hulp@salud.madrid.org) to request access approval.

**Funding:** This study was funded by an unrestricted grant from Abbvie, which was fully invested in improving the digital solution. The funders had no role in study design, data collection and analysis, decision to publish, or preparation of the manuscript.

**Competing interests:** Diego Benavent has received speaker fees from Jannsen, Roche, grant/research support from Novartis and Abbvie. Francisco J. Núñez-Benjumea was an employee of AdheraHealth Inc at the time this research was conducted. Luis Fernández-Luque is an employee of AdheraHealth Inc. Victoria Navarro-Compán has received speakers fees from AbbVie, Eli Lilly, Janssen, MSD, Novartis, Pfizer, UCB Pharma; she has been consultant of: AbbVie, Eli Lilly, MSD, Novartis, Pfizer, UCB Pharma; received grant/research support from AbbVie and Novartis. María Sanz: None declared. Marta Novella-Navarro reports grants from UCB, Lilly and Janssen. Enrique Calvo-Aranda has received speaker fees from Abbvie. Leticia Lojo: None declared. Alejandro Balsa has received speaker fees from Pfizer, Abbvie, Lilly, Galapagos, BMS, Sandoz, Nordic Pharma, Gebro, Roche, Sanofi, UCB; he has been consultant of Pfizer, Abbvie, Lilly, Galapagos, BMS, Nordic Pharma, Sanofi, UCB; received grant/research support from Pfizer, Abbvie, BMS, Nordic Pharma, Gebro, Roche, UCB. Chamaida Plasencia has received speaker fees from Pfizer, Abbvie, Lilly, Sandoz, Sanofi, Biogen, Roche, Novartis; received grant/research support from Pfizer and Abbvie.

## Author summary

There has been an exponential growth in the use of communication technologies to support diagnosis, monitoring or treatment in recent years. In rheumatology, the use of mobile applications, provides an opportunity to improve disease management through the collection of large amounts of data. This can be done using electronic patient-reported-outcomes (ePROs). Here, we present the development and implementation of a mobile solution and a hybrid care model. We found the use of ePROs feasible to monitor disease activity, flares and problems with the medication in patients with rheumatoid arthritis (RA) and spondyloarthritis (SpA). Our study provides new insights into the use of digital health technology in rheumatology, highlighting the importance of the involvement of both healthcare professionals and patients the implementation of mobile health in clinical practice

## Background

Rheumatic and musculoskeletal diseases (RMDs), a heterogeneous group of diseases that affect joints, bones, muscles, tendons and ligaments, have increasing importance due to their prevalence and impact on patients' lives. Hundreds of thousands to millions of people worldwide are affected by such RMDs as rheumatoid arthritis (RA) and spondyloarthritis (SpA), with global prevalence estimates of 0.46% (95% confidence interval [CI] 0.39–0.54) and 0.2%-1.6%, respectively [1,2]. Clinical management of these diseases requires tight monitoring of physical and psychological symptoms, which is not always feasible in an outpatient setting due to the time constraints and burden of care [3]. For these reasons, telemedicine and mobile health (m-Health) technology represents as a novel opportunity for the clinical management of chronic patients.

Electronic patient-reported outcome measures (ePROs) may be captured using mHealth tools, facilitating insight into symptoms that occur in between visits [4]. In addition, mHealth can also allow for the monitoring of symptoms that might be overlooked by rheumatologists in time-constrained visits. This can help reduce the mismatches in issues important to both patients and clinicians [5]. Digital solutions and wearables can empower patients by supporting self-management skills and/or providing feedback on health behaviour or symptoms management [6].

However, both the development and implementation of mHealth technologies present several challenges and risks. Some recent publications have shown that the quality of most rheumatology apps is not high, with several attendant and important pitfalls [7,8]. This lack of high-quality apps highlighted the need for some standardization in their development, in order to ensure patient safety and enhance the usefulness of this technology. In this regard, the European Alliance of Associations for Rheumatology (EULAR) published the EULAR points-to-consider (PtC) guidelines for the development, evaluation and implementation of apps aiding self-management of RMDs [9]. These points provided a basis for the development of future mHealth solutions to aid the self-management of people living with RMDs, and further contribute to existing self-assessment tools.

In this context, the Digireuma study (ISRCTN11896540) included the design of the Mixed Attention Model (MAM), designed to facilitate close monitoring of patients while providing a tailored intervention to improve self-management skills. The MAM is a care model implemented in clinical practice that incorporates the use of information and communication

technologies (ICT) in patient follow-up. It enables patient monitoring and education, thereby facilitating more effective and efficient management than traditional methods of patient follow-up. A Precision Digital Companion Platform, Adhera for Rheumatology, was adapted for enabling the implementation of the MAM.

The objective of this feasibility study was to test the use of ePROs to monitor disease activity, functionality, overall health, flares and problems with the medication as well as to evaluate user experience of patients with RA and SpA with this digital health solution.

## Methods

The Digireuma study involved four successive stages **Fig 1**. First, the unmet needs as well as the potential benefits of using digital solutions for patients with RMDs in Spain were evaluated through a focus group meeting. Subsequently, a hybrid (virtual and face-to-face) intervention strategy was conceptualized and later registered as a protocol (ISBN:978-84-09-38502-7). Following the content development of the care model, a digital platform was technologically adapted. Finally, a feasibility study was conducted to evaluate its implementation.

### Phases in the study design

**Phase I: Identification of unmet needs—focus group.** Since 2000, the Complex Therapy Unit (CTU) of the Hospital Universitario La Paz (HULP) has attended 1,563 patients with RMDs treated with biologic or target specific disease-modifying anti-rheumatic drugs (b/tsDMARDs), of whom 842 are currently under active therapy. In the CTU, all patients under b/tsDMARDs are clinically evaluated by rheumatologists with a regimen of protocolized visits. This number is progressively increasing due to the chronic nature of these diseases and to the therapeutic strategies involving stricter outcome measures. Worth noting is the fact that the current care models only provide information on disease evolution obtained during the established clinical visits, but no real data for the intermediate periods. This means that relevant information about the evolution of the patient's disease such as flares, incidents with treatment or infections, can be lost due to patients' recall bias. On the other hand, CTU patients have different degrees of disease activity and not all require the same regimen of face-to-face visits.

Healthcare professionals from the CTU had raised these concerns and leading to the formation of a focus group of patients with RA. To establish the focus group, 20 patients with RA with a recent clinical visit were contacted by telephone in September 2019. The purpose of the meeting was explained to them, and only five patients were available to attend a face-to-face meeting that lasted 120 minutes. Patients with different ages were involved in this focus group. Only patients with RA were included because this pilot project was initially addressed to this disease. Later, in the study design, patients with SpA were also included because both are the most frequent pathologies under b/tsDMARDs in our unit.

Two clinicians (DB, ChPR) developed the focus group guidelines and led the process. The clinicians facilitated the group with the aim to explore the unmet needs of patients, as well as their use of technology and preferences regarding mHealth. An information sheet and a pre-survey was sent to the patients ahead of the meeting to explain the purpose of the study (**S1 Text**). A structured questionnaire with open questions about the use of mHealth were developed for the focus group to explore the perceived usefulness and benefits of apps. Afterwards, the different proposed features were ordered by the participants on a scale from 1 to 10 (higher scores indicating more importance in an mHealth solution). The interviews not only explored the preferred methods of remote monitoring, but also helped define which ePROs might be collected during the project and the frequency of such collection. The focus group meeting was held at HULP. Responses of the participants were transcribed and analyzed. The themes

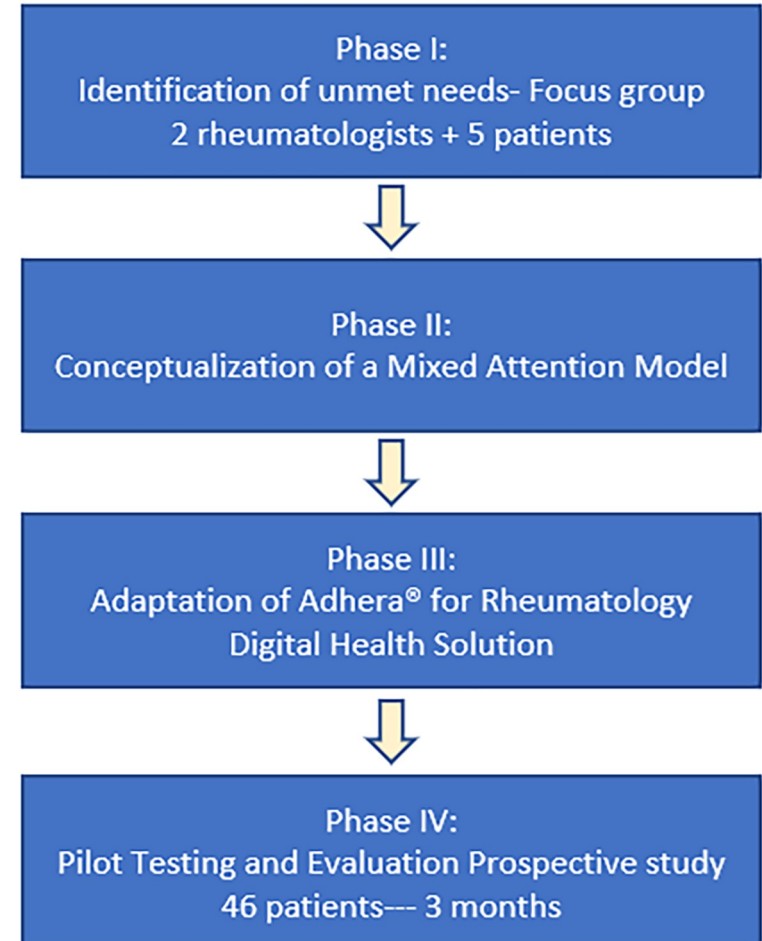

**Fig 1. Phases of design and implementation of the Digireuma study.**

for the analysis of the transcriptions aimed to identify key patient needs and help define the set of ePROs to be incorporated into a digital solution. Patients gave their consent to use the interview materials for research purposes, and to publish the relevant data.

**Phase II: Conceptualization of protocol with a Mixed Attention Model (MAM).** The focus group meeting led to the development of a protocol that combines asynchronic virtual and face-to-face monitoring in patients with rheumatic diseases, which was designated the Mixed Attention Model (MAM). With this model, patient follow-up combines traditional face-to-face visits with self-home-monitoring simultaneously. MAM was further adapted into a protocol applicable to clinical practice whereby the flow of procedures in face-to-face visits and virtual follow-ups were explained in detail elsewhere (ISBN:978-84-09-38502-7).

Patients were provided with access to a digital solution that can be downloaded free of charge and whose design was developed and supervised by the professionals involved in the CTU. Patients could then complete questionnaires addressing activity, disability, and other domains as patient-reported outcomes (PROs) and obtain information about their pathology and other interesting aspects impacting their daily lives. The incidents recorded in the MAM were periodically reviewed via a web interface by a clinician partly dedicated to this purpose (digital clinician). This professional, or another from the CTU (on-site clinician), contacted

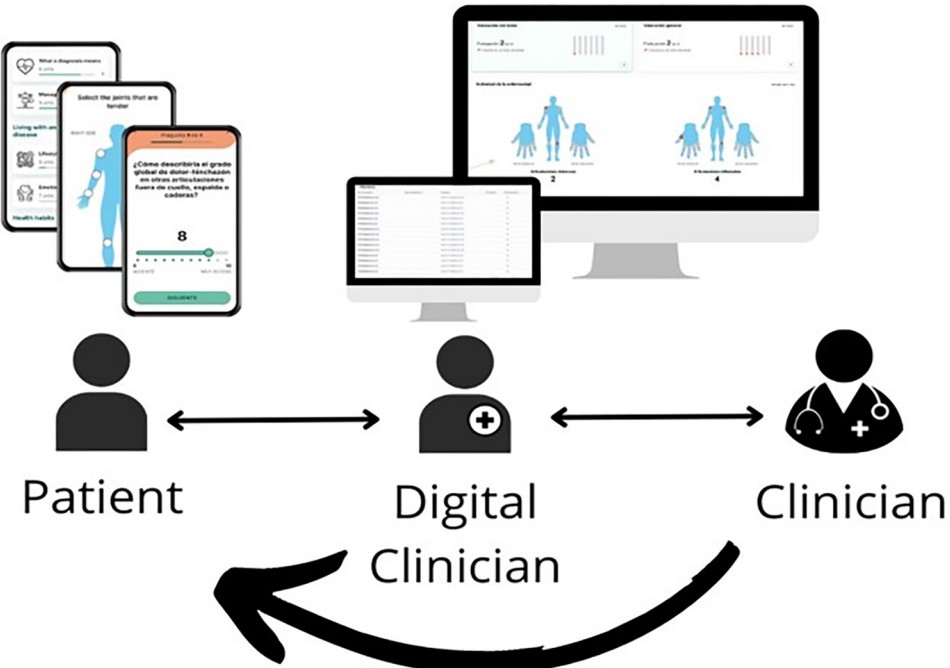

**Fig 2. Digital monitoring in the study powered by Adhera for Rheumatology.** Screenshots at top depict the mobile interface (left) and the clinical web application (right).

the patient by telephone to resolve the reported incident (*Fig 2*). The alert could be resolved entirely remotely, or with a face-to-face visit.

**Phase III: Conceptualization-Adaptation of a Digital Health Solution (Adhera).** In this study, we adapted the Adhera for Rheumatology digital program, which was powered by the Adhera Digital Precision Digital Companion Platform. The platform contains modules for personalized education, behavioral change and mental wellbeing that have been already used across multiple studies and implementations across different therapeutic areas [10–12]. The adaptation of the digital intervention program for this study by a multidisciplinary team included rheumatologists, a general physician, psychologists, and digital health specialists. It could be downloaded to patients' mobile devices; they could then complete, or consult, the content at home according to their preferences or needs.

This mHealth solution included ePROs related to the disease and disease-specific educational content, as well as a set of functionalities to promote self-management skills. Thus, motivational messages were delivered via an artificial intelligence-based health recommender system, depending on the rating of previous messages and the results of ePROs. These motivational messages addressed behavioral and self-management aspects, such as goal setting for physical activity. As an example of these: «*How are you? To have a clear objective can help you to be more active. Set up an objective for exercising including how long or how many steps, when and the intensity you will do it. For example, "I will do brisk walking Tuesdays and Thursdays 30 minutes*" ». In addition, a set of mental well-being activities were made available to the user, including mindfulness exercises and relaxation techniques. The digital solution notified the patient of pending questionnaires and sent motivational messages. Furthermore, flares and changes in medication could be reported at any time by the patient. This information was transferred to a web interface that was reviewed by the digital clinician at least two times a week.

In regards to the included ePROs, clinicians (with the input of patients) agreed on a set commonly used in clinical practice. This defined set was incorporated in Adhera for Rheumatology and were completed by patients at home. Assessment measures were delivered to patients with a pre-established time frequency (as assessed in the focus group and including rheumatologist input) through the digital platform, and patients were informed of the possibility of completing the outcomes at that predefined time. Additionally, patients had the possibility of reporting a flare when they deemed it necessary. For patients with RA, the following outcomes were included: patient global assessment for disease activity (PGA), self-assessed tender joint count (s-TJC), self-assessed swollen joint count (s-SJC), Health Assessment Questionnaire (HAQ) and visual analogue scale (VAS); meanwhile, for patients with SpA, the following assessments were included: VAS, PGA, s-TJC, s-SJC, Bath Ankylosing Spondylitis Disease Activity Index (BASDAI), and ASAS Health Index (ASAS-HI). In addition, problems addressing medication were queried. The frequency with which these tools were made available for completion was previously established by consensus of the multidisciplinary team involved in the design. (**S1 Table**).

The educational content and functionalities to promote self-management skills were made available to patients for consultation and completion according to the preferences and availability of the patients, without requiring any pre-established frequency. On the other hand, the personalized motivational messages were designed by professionals and reviewed by clinicians, mainly focused on self-management and disease coping techniques.

The design of the application encompassed concerns regarding data protection for both patients and investigators. A secure server was used to store patient data, and patients were pseudonymized in accordance with data protection laws. Due to research protocol privacy constraints, it was not possible to permit the entry of personal emails into the mobile solution, and therefore users were provided with a user-id, tailored to their disease (i.e., RA or SpA).

**Phase IV: Digireuma prospective study.** In order to evaluate the impact, use, and feasibility of Adhera for Rheumatology in routine care, patients were consecutively recruited during a period of one month from our CTU. Inclusion criteria were: patient age above 18 years, diagnosis of RA or SpA by their treating rheumatologist, treatment with an advanced therapy (b/tsDMARDss) for at least one year and smartphone availability. The exclusion criteria were: any conditions that hindered or prevented the use of a mobile application (blindness, dementia, illiteracy, etc.). Patients were contacted and invited to the project, until the number of patients considered optimal for this feasibility project (at least 20 patients per disease) was reached. Sample size was not based on power calculations because there are few reliable estimates in the literature regarding digital health solutions. Hence, due to the exploratory character of the study, no sample size calculation was performed. To evaluate the representativeness of the study cohort as a general RA and SpA population in terms of rheumatologic specialized care, the characteristics of the patients in the Digireuma study were compared to the baseline characteristics of a sample cohort from the CTU, matched by disease.

After signing informed consent forms, patients were included in the study. Small groups of up to 10 patients were gathered on different days in a hospital assembly hall where the study and the digital solution were explained by two clinicians. In those cases in which a patient could not attend a meeting, a personal report was conducted instead, either online or face-to-face. Following a tailored training program, patients downloaded the digital solution and completed a paper-based baseline visit. A user-id for the Adhera for Rheumatology Precision Digital Companion Platform was provided to each patient according to their disease, either RA or SpA. Two rheumatologists monitored these ePROs and patients were contacted for online or face-to-face interventions when deemed necessary by the clinicians (**Fig 2**). Of note, one of

them was in charge of reviewing the incidents remotely, while the other clinician carried out the telematic or face-to-face consultation.

Patients were invited to participate for a maximum of 6 months of follow-up, but for this initial feasibility study, data were analyzed 3 months after recruitment. Follow-up consisted of clinical monitoring according to the MAM, combining traditional face-to-face visits and continuous digital follow-up through Adhera for Rheumatology. On the one hand, patients completed ePROs remotely to provide clinicians with information about their health status. Moreover, incidents recorded in the tool were periodically reviewed on a web interface by the digital clinician, a rheumatologist whose clinical work was partly dedicated to this purpose. If deemed necessary, patients were contacted by telephone to resolve any reported incidents. These could be resolved entirely remotely, or via a face-to-face visit with the on-site clinician. On the other hand, patients were followed-up via face-to-face visits in accordance with our department's standard procedures (i.e., approximately at 3-month visits).

Number of interactions, including questionnaire items, accesses to educational units, quizzes and rated messages were assessed. Additionally, alerts during the follow-up period were evaluated. Regarding the assessment of the digital solution, the usability of Adhera for Rheumatology was measured making use of the Net-Promoter Score (NPS). The subjective quality of the application was measured using the Mobile Application Rating Scale (MARS) item [13]. In particular, we asked "What is your general evaluation of the application?" which included a 5-start rating (one being the worse and 5 the best). Moreover, the feasibility and clinical relevance of the process was assessed at the end of the three-month follow-up by semi-structured telephone interviews conducted by the clinicians (DB, ChPR) to all recruited patients. These aimed to assess patients' satisfaction as well as any relevant issues on the use of the mobile solution.

## Statistical analysis

This feasibility study includes data from baseline through 3 months. The statistical analysis was conducted using SPSS version 24 to analyze the demographic, clinical and patient reported variables. Results are presented as relative frequencies for categorical variables and median and interquartile range (Q1, Q3) for continuous variables. Pearson chi-squared and Fisher exact tests were used to compare the frequency data between groups. Mann-Whitney U and Wilcoxon nonparametric tests were used to compare the continuous data. Values of $p < 0.05$ were considered statistically significant.

## Ethical considerations

The local ethics committee from Hospital Universitario La Paz approved the study (PI-4519) and Digireuma was registered with the ISRCTN Registry (ISRCTN11896540). All patients signed informed consent.

## Results

### Focus group

The focus group included 5 patients with RA, among whom 2 were female. Mean age was 62 ± 16.7 years, mean disease duration 16.8 ± 13 years, and mean time under biological drug 7.3 ± 5.6 years. Four out of five patients acknowledged using apps frequently. Patients suggested potentially helpful features to meet their unmet needs for managing their disease. These features, as well as agreement among the entire group on their relative importance and the final output in Adhera for Rheumatology are detailed in Fig 3. In the open discussion, several

| Recommended app feature | Relevance to patients | Final output in Adhera Rheumatology Digital Program |
|---|---|---|
| Urgencies manager | | Flare alarm |
| Consultation reminders | | Personalized calendar* |
| Blood tests organizer | + | Personalized calendar* |
| Communication with physicians | | Alarms: flare, medications. |
| Treatment manager and reminders | | Personalized calendar* |
| Information about the disease | | General information about the pathologies, healthy habits, and frequently asked questions |
| Self-monitoring of health outcomes | | Tailored ePROs for each disease |
| Side effects notifications | | Medications (side effects) alarm |
| Rehabilitation exercises | - | Non-pharmacological recommendations and mindfulness exercises |

*A personalized calendar was created, but not implemented, in the first phase of Digireuma

**Fig 3. Features requested by patients, by relevance, in the focus group as well as by final output included Adhera Rheumatology Digital Program.**

patients' comments revolved around the importance of an appointment reminder and the opportunity to remotely assess their lab test results. They also assigned importance to remote monitoring with measurements concerning their health status, enabling the ability to update the physician more frequently. Regarding usability, the greatest emphasis was placed on having an application that was simple to use and with well-defined functions. Patients insisted on the importance of the physician being able to use the data entered in the tool to understand the patient's condition more precisely (either by direct connection with the hospital or by taking their cell phone to the consultation).

## Pilot-testing study

From a cohort of 842 patients, 73 were contacted in May 2021, and 46 signed inform consent and were therefore provided access to the MAM and Adhera for Rheumatology platforms; however, only 41 patients completed the onboarding and actively used Adhera for Rheumatology (**S1 Fig**). Of the 46 included patients, 22 had RA and 24 SpA. Mean age was 48± 12 and 42 ± 9 years in the RA and SpA groups, respectively. Eighteen of twenty-two (82%) patients with RA were female, whereas 9/24 (38%) with SpA were female. Among RA patients, 2/22 (10%) presented rheumatoid nodules, 1/22 (5%) interstitial lung disease, 1/22 (5%) Sjögren syndrome and 1/22 (5%) autoimmune neutropenia. Out of the 23 patients with SpA, two were considered purely axial SpA, two purely peripheral SpA, and 19 have both axial and peripheral involvement. Concerning extraarticular manifestations in SpA, 4/24 (16.7%) presented uveitis, 2/24 (8.3%) psoriasis and 2/24 (8.3%) inflammatory bowel disease. **Table 1** shows clinical characteristics of the included patients stratified by disease at digital solution baseline.

**S2 Table** shows characteristics of patients included in this pilot study as compared with a group of patients from the CTU. In the RA group, there were no differences between patients in Digireuma and the general cohort for certain characteristics, such as gender, smoking habit, physician global assessment (PhyGA) or patient global assessment (PtGA). Nevertheless, the two groups differed in some relevant characteristics. For example, those in Digireuma were younger, presented worse DAS-28, C-reactive protein (CRP) and Health Assessment Questionnaire (HAQ) results. In the SpA group, there were no differences between groups in terms of gender, HLA*B27 positivity, or any of the assessed outcomes, including BASDAI, PhyGA, PtGA and CRP. However, patients included with SpA in Digireuma were younger.

Among the included patients in the Digireuma prospective study, 41 (89%) completed the onboarding [18/22 (82%) RA, 23/24 (96%) SpA] and 37 (80%) submitted at least one entry.

**Table 1. Clinical characteristics stratified by disease at digital solution baseline.**

|  | RA patients (n = 22) | SpA patients (n = 24) |
|---|---|---|
| **Demographic and clinical features** |  |  |
| **Sex (female)** | 18 (81.8) | 10 (41.7) |
| **Age (years)** | 47.4 (42.3, 54.3) | 40.7 (38.7, 44.9) |
| **Smoking habit (ever smoker)** | 12 (54.5) | 9 (38.1) |
| **RF positive** | 14 (63.6) |  |
| **ACPA positive** | 16 (72.7) |  |
| **HLA*B27 positive** |  | 13 (68.4) |
| **Baseline measurements** |  |  |
| **DAS-28** | 2.9 (1.8, 3.9) |  |
| **HAQ** | 4.5 (0, 8.3) |  |
| **BASDAI** |  | 2.1 (1.2, 3.5) |
| **PhyGA** | 25.0 (10.0, 50.0) | 10.0 (5.0, 20.0) |
| **PtGA** | 25.0 (10.0, 42.5) | 20.0 (10.0, 45.0) |
| **CRP (mg/L)** | 0.6 (0.5, 1.5) | 0.5 (0.5, 2.3) |
| **Concomitant treatment** |  |  |
| **csDMARDs** | 19 (86.4) | 11 (50.0) |
| **Prednisone** | 10 (45.5) | 0 |

**Results are expressed in median (Q1, Q3) or n (%).** ACPA: anti-citrullinated peptide antibodies; RF: rheumatoid factor; CRP: C-Reactive Protein; BASDAI: Bath Ankylosing Spondylitis Disease Activity Index; PhyGA: physician global assessment; PtGA: patient global assessment; csDMARDs: conventional synthetic disease-modifying antirheumatic drugs.

Five patients fell out of inclusion because they did not download the digital solution. There was a total of 7,179 interactions during the first three months of follow-up. In the RA group there were a total of 4,019 total interactions (2,178 questionnaire items, 648 accesses to educational units, 105 quizzes and 1,088 rated messages), while patients with SpA (n = 23) had a total of 3,160 interactions (1637 questionnaire items, 684 accesses to educational units, 77 quizzes, 762 rated messages). **Table 2** shows ePROs measurements completion rates for RA and SpA patients who completed any data during follow-up. Patients with RA completed a median of 9.5 (4.3, 15.8) ePROs during the 3-months follow-up, with HAQ the ePRO that was answered by the greatest number of patients (n = 16). Likewise, patients with SpA completed a median of 3 ePROs (1, 12). Concerning alerts, 15 patients generated a total of 26 alerts, of

**Table 2. Onboarded patient engagement with regards to ePROs.**

| Rheumatoid Arthritis (n = 18) |  |  |  |  |  |  |
|---|---|---|---|---|---|---|
|  | PtGA | s-TJC | s-SJC | VAS | HAQ | Total |
| ePROs completed | 1.5 (0.25, 3) | 2 (0.25, 3) | 2 (0.25, 3) | 2 (0, 3) | 2 (1, 3) | 9.5 (4.3, 15.8) |
| Patients with ≥ 1 entry | 13 (72.2) | 13 (72.2) | 13 (72.2) | 12 (66.7) | 16 (88.9) | 16 (88.9) |
| **Spondyloarthritis (n = 23)** |  |  |  |  |  |  |
|  | PtGA | s-TJC | s-SJC | BASDAI | ASAS-HI | Total |
| ePROs completed | 1 (0,3) | 1 (0,3) | 1 (0,3) | 1 (0,2) | 1 (0,2) | 3 (1, 12) |
| Patients with ≥ 1 entry | 16 (69.5) | 16 (69.5) | 16 (69.5) | 14 (60.8) | 14 (60.8) | 21 (91.3) |

Follow-up period was 3 months. Results are expressed in median (Q1, Q3) and n (%). PtGA: Patient Global Assessment of disease activity; s-TJC: self-assessed Tender Joint Count; s-SJC: self-assessed Swollen Joint Count; VAS: Visual Analogue Scale; HAQ: Health Assessment Questionnaire; BASDAI: Bath Ankylosing Spondylitis Disease Activity Index; ASAS HI: Assessment of SpondyloArthritis International Society Health Index

which 24 were flares (10 RA, 14 SpA) and 2 were problems with the medication, 1 RA and 1 SpA). Eighteen (69%) of the alerts were solved remotely, 5 (19%) required a face-to-face intervention and 3 (12%) patients did not respond prior to the face-to-face consultation.

The NPS score was administered to the patients via the digital solution to assess the perceived satisfaction and usability. According to the NPS, 9/14 were considered promoters, 4/14 passives and 1/14 was a detractor, resulting in a NPS score of 57. The general rating of the app was 4.3/5. All 46 patients were contacted by telephone to make a semi-structured interview at 3 months, aiming to obtain feedback from patients and identify incidences with the digital solutions that could be improved by the research team. Only in one patient was not possible to contact despite several attempts. Eight patients (4 RA and 4 SpA) had technical problems because the automatic password recovery process failed, then they could not use the digital solution. In general, the remaining contacted patients (35 in total) expressed satisfaction with the initiative of the project and with the digital solution. The negative aspects most frequently reported were motivational messages (repetitive or not suitable) and reminders of PROs (the lack of feedback after completing them when disease is stable). The recurrent positive comments were having the possibility to inform the professional about incidents/evolution between clinical visits and having a source with reliable information about their disease. Among the suggestions for future improvements, the benefit of adding a calendar that included patient-tailored information and a more direct messaging tool for contacting healthcare professionals (not only for reporting flares) were highlighted.

## Discussion

The Digireuma project consisted of the design and pilot-testing of a digital health solution to monitor patients with RA and SpA and enable them to report flares and track PROs. Adhera for Rheumatology allowed patients with RA and SpA to integrate ePROs into their daily routine, providing clinicians with assessments of disease activity and functional status. Digital monitoring through ePROs was shown as a feasible alternative to standard care and PRO completion in paper format. During the prospective 3-month follow up, 90% patients completed some data using the digital solution, with more than 7,000 interactions, of which 26 lead to an intervention (18 of them [70%] remote). Concerning usability and satisfaction, although only 14 patients rated the solution, the majority of them (9 patients [64%]) were considered promoters.

The current recommended methodology to develop mHealth solutions for RMDs patients was recently published in the EULAR PtC for the creation, evaluation, and implementation of mobile health applications [9]. These PtC provide guidance on important aspects on mHealth which relate to patient safety, relevance of the content and functionalities, transparency, involvement of health care professionals, usability and cost-benefit balance. More recently, the EULAR PtC for the development, prioritisation and implementation of telehealth for people with RMD have also been developed, identifying areas where telehealth may improve quality of care and increase healthcare access [14]. These principles have been addressed throughout the development, evaluation, and implementation of the MAM and Adhera for Rheumatology platform. This was achieved by first taking into account patients' needs and requirements though the development of a focus group to address safety, relevance of the content and functionalities, and usability. Second, the involvement of healthcare professionals was made central to the project at the very outset, with the co-creation of a platform involving clinicians, researchers, digital health experts and technicians. Transparency has been maintained throughout by the developers, to include the funding source, content validation process, and version updates; additionally, data ownership, and compliance with regulatory frameworks has been considered by the research team along the process [15]. The cost-effectiveness of the

fully implemented solution will be further assessed by determining the number of consultations that were avoided by those patients who were followed up by MAM.

There is growing evidence that ePROs are a reliable tool for monitoring patients, since they allow for the capture of information relevant to patient management, such as disease activity [16]. In a recent study, the acquisition of ePROs was shown to be a reasonable alternative to paper-pencil formats in terms of equivalent outcomes in the app versus paper-based method, in addition to being preferred by the participants [17]. Indeed, the transmission and management of data through mHealth solutions not only helps optimize clinical visits, but also enables investigations to be conducted remotely from the researcher's location. However, mHealth interventions might not be suitable for all patients, and specific patient profiles that might most benefit from mHealth interventions has yet to be defined. Hence, established routine visits were maintained in this pilot study, while determining which patients might benefit most from such face-to-face visits remains one of the main goals of the project.

Despite the surge of ePROs and mHealth, development and implementation of mHealth solutions in rheumatology is still very scarce. Unlike other areas of medicine, such as mental health, the availability of robust evidence in mHealth for RMDs is lacking [13]. One of the most robust projects to date is the REmote MOnitoring of RA (REMORA) study, that aimed to test the feasibility of collecting daily patient-reported symptoms from 20 RA patients over three months through an app, being integrated in the electronic health record [18]. Seppen et al developed a twelve month, randomized clinical trial in RA patients to assess non-inferiority in terms of ΔDAS-28 after 12 months and the ratio of the mean number of consultations with rheumatologists as compared to usual care patients. The study included 103 patients with stable low disease activity, and it showed that, in terms of DAS-28, patient care backed by smartphone self-monitoring was comparable to standard care and reduced rheumatologist visits by 38%[19]. According to a recent review that sought to identify the currently available health information technology tools for collecting ePROs in RMDs, only 10 such tools were identified, most of them being web-based [20]. Out of these 10, 9 were for RA, whereas only 1 was for SpA. None of the tools could be adapted to patients with RA and SpA. Therefore, to our knowledge, Adhera for Rheumatology is the first digital platform that allows for the telemonitoring of patients with RA and SpA.

Collaboration between healthcare professionals and patients was essential for the development of Adhera for Rheumatology. In line with this, a previous study aiming to explore the needs, experiences, and views of people diagnosed with RMDs using mHealth apps revealed that patients believed a mobile solution could help them manage their condition if it was co-developed with healthcare professionals [21]. Interestingly, patients were mainly interested in mHealth solutions that enabled them to self-monitor their health parameters or to communicate directly with their health care provider. Overall, these conclusions are in accordance with the comments reported in our own study's focus group. An attractive proposition for patients would be to integrate clinician support into their daily lives, thereby providing access to healthcare expertise at the precise time it is needed. Thus, disease and socio-cultural factors may play an important role in the use of digital technologies; younger patients, with a long-standing diagnosis, and with stable disease activity might be more prone to use digital solutions. In this sense, there is a need to explore the quantification of engagement in digital interventions. Social determinants of health are undoubtedly in this future research agenda. Hence, age should be considered since some RMDs may present in older patients. Perceptions and resistances on the use of telemedicine should be explored. Geography is a critical factor; although in our environment hospitals are accessible, many patients in some other areas may have mobility restrictions so there is a greater need for telemedicine implementation. One of the strengths of this study is that there was a meeting with a focus group of patients to identify

whether the population was interested in digital tools and what unmet needs stood out in the follow-up. The mean age of the patients in the focus group was higher than that of the patients in the pilot study, which can be explained by one patient with outlier older age in the focus group which affected the mean, together with a general trend of older participants in the group due to logistics; as the meeting was face-to-face, the availability to attend in younger people due to job constraints was higher. In this sense, patients with SpA were not included in the assessment of the focus group. Patient profiles may also have had an impact on the pilot study. Due to the intrinsic characteristics of the disease, patients with SpA involved had a younger profile, with greater proportion of men, as compared with these with RA, which may have influenced their engagement in the solution. Future research should further investigate how social determinants of health impact the use of digital solutions in rheumatology. Besides, we would like to explore the quantification of engagement in digital interventions that include both an element of ePROs/monitoring and patient empowerment/education. In such case, we can foresee that engagement can also lead to lower symptom burden and potentially low interest in reporting ePROs on symptoms. Additional evidence must be gathered to determine the best strategy for implementing ePROs for patients with RMDs.

Since this was the first stage of a pilot project, there were a few barriers that hampered the inclusion of more patients. Some of the contacted patients were reluctant to participate; 73 were contacted and 46 signed inform consent. In fact, a lower age of the patients enrolled in the study as compared to the standard care group indicates a potential reluctancy of older patients in the participation, indicating age as a barrier for the implementation of new technologies. In addition, another aspect that had a negative influence on recruitment was a potential fear of patients that their face-to-face visits will be replaced by telematic visits. COVID-19 pandemic might have been also a driver to adoption of the intervention; in one hand, some patients were more familiarized with technology; however, others might have been afraid that technology has been discussed as an alternative to face-to-face consultation. Hence, a potential cultural barrier on the use of these new technologies may have play a role in the number of patients participating in the study. Besides, human workforce who could devote time for this project was limited, both in the clinical and technical part. On top of this, clinicians have not received formal training on the implementation of these technologies, which made it more difficult to address the challenges along the project. Nonetheless, this study was not designed to gather insights on those aspects.

While the Digireuma study is a promising beginning towards incorporating remote monitoring into rheumatology care, certain limitations must be acknowledged. Regarding development of a digital solution, not all of the features deemed important to patients could be integrated into the digital solution at this stage. For example, patients mentioned the usefulness of features that would remind them of their medications, appointments and also provide access to test results. These features were not included in Adhera for Rheumatology since it would have entailed the development as a medical device, which was outside the scope of the project. Moreover, it was not integrated into the hospital information system. Nevertheless, Adhera for Rheumatology allowed remote monitoring with measurements of health status, which patients also found beneficial. Besides, patients with SpA were not included in the assessment of the focus group. Although clinicians involved in the study have a specialization in SpA and assessed the pertinence of ePROs in this group, a less adequate of specific ePRO may have been a reason why the adherence in this group was lower than in RA. Patients in the Digireuma study presented some differences with those in the standard care population. Despite patients were selected randomly from clinical practice to participate in Digireuma, it is plausible that younger patients were more prone to participate in the study, with older patients declining participation, biasing the selection of patients. Further studies should look into the

acceptance of the digital intervention across different demographics, especially those that might pose a health disparity. Furthermore, patient engagement with the app proved challenging throughout the project. Whereas 4 of 5 patients transmitted at least some data during follow-up, there were few who completed most of the assessments and user engagement was lower than expected. This was mainly due to problems with the login process, which stemmed from research protocol privacy constraints preventing the entry of personal emails into the mobile application for password recovery purposes. Besides, different patient-reported outcomes were displayed periodically and not accessed at once by patients, which hampers the calculations of disease activity indices, such as DAS-28 or ASDAS. Furthermore, a significant discrepancy between self-reported and physician-measured arthritis has been reported in previous studies [22], which may limit the validity of the joint-count results. This might also indicate an opportunity for health literacy interventions to support the self-monitoring by patients. As for patient satisfaction, the NPS, which provides valuable information on patient opinion and improvements to be implemented, was only completed by 34% of patients. For later stages of the project, more active intervention, including a phone call in lieu of using the app, is planned. As an additional limitation, our study was conducted at a single tertiary center, precluding any external feasibility assessment of the digital solution in different healthcare settings. However, during the next phases of the study another hospital will also recruit patients, thereby affording a multicenter setting. In our future agenda, we have to analyze the data in patients who completed 6 months of follow-up. Also, the correlation between data reported remotely with data from protocolized face-to-face visits will be studied.

In conclusion, this study shows that the use of a digital health solution is feasible in clinical practice. The involvement of both healthcare professionals and patients is critical for mHealth implementation in clinical practice. Based on these preliminary results, the next step will be to further implement the Precision Digital Companion Platform, Adhera for Rheumatology, in a multicentric setting to analyze the additional benefits of this patient monitoring system and to assess the results of the implementation of the MAM in a clinical setting.

## Supporting information

**S1 Text. Patients' survey (focus group).**
(DOCX)

**S1 Fig. Flowchart of the prospective study.**
(PNG)

**S1 Table. Frequency of assessment of ePROs.**
(DOCX)

**S2 Table. Clinical characteristics of patients in Digireuma as compared with standard of care.**
(DOCX)

## Acknowledgments

The authors thank the Spanish Foundation of Rheumatology for providing editorial assistance during the preparation of the manuscript (FERBT2022)

## Author Contributions

**Conceptualization:** Diego Benavent, Luis Fernández-Luque, Francisco J. Núñez-Benjumea, Alejandro Balsa, Chamaida Plasencia-Rodríguez.

**Data curation:** María Sanz-Jardón, Chamaida Plasencia-Rodríguez.

**Formal analysis:** Diego Benavent, Francisco J. Núñez-Benjumea.

**Funding acquisition:** Diego Benavent.

**Investigation:** Diego Benavent, Victoria Navarro-Compán, María Sanz-Jardón, Chamaida Plasencia-Rodríguez.

**Methodology:** Diego Benavent, Luis Fernández-Luque, Francisco J. Núñez-Benjumea, Victoria Navarro-Compán, Marta Novella-Navarro, Alejandro Balsa, Chamaida Plasencia-Rodríguez.

**Project administration:** Diego Benavent, Luis Fernández-Luque, Francisco J. Núñez-Benjumea, Pedro L. González-Sanz, Chamaida Plasencia-Rodríguez.

**Resources:** Luis Fernández-Luque, Marta Novella-Navarro, Pedro L. González-Sanz, Enrique Calvo-Aranda, Leticia Lojo, Chamaida Plasencia-Rodríguez.

**Software:** Luis Fernández-Luque, Francisco J. Núñez-Benjumea.

**Supervision:** Luis Fernández-Luque, Victoria Navarro-Compán, Alejandro Balsa, Chamaida Plasencia-Rodríguez.

**Validation:** Victoria Navarro-Compán, Marta Novella-Navarro, Chamaida Plasencia-Rodríguez.

**Visualization:** Diego Benavent.

**Writing – original draft:** Diego Benavent.

**Writing – review & editing:** Diego Benavent, Luis Fernández-Luque, Francisco J. Núñez-Benjumea, Victoria Navarro-Compán, María Sanz-Jardón, Pedro L. González-Sanz, Leticia Lojo, Alejandro Balsa, Chamaida Plasencia-Rodríguez.

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
