## [Decision Letter · Decision Letter 0]

25 Jul 2022

PDIG-D-22-00071

Monitoring Chronic Inflammatory Musculoskeletal Diseases Mixing Virtual and Face-to-Face Assessments - Results of the Digireuma Study

PLOS Digital Health

Dear Dr. Benavent,

Thank you for submitting your manuscript to PLOS Digital Health. After careful consideration, we feel that it has merit but does not fully meet PLOS Digital Health's publication criteria as it currently stands. Therefore, we invite you to submit a revised version of the manuscript that addresses the points raised during the review process. Please ensure you address the reviewer comments related to the additional detail and clarifications requested in the background, methodology, results and discussion sections and clarification of the data presented in the tables and supplementary information. 

Please submit your revised manuscript within 60 days Sep 23 2022 11:59PM. If you will need more time than this to complete your revisions, please reply to this message or contact the journal office at digitalhealth@plos.org. Please include the following items when submitting your revised manuscript:

We look forward to receiving your revised manuscript.

Kind regards,

Belinda Lange

Academic Editor

PLOS Digital Health

Journal Requirements:

Additional Editor Comments:

Please review the manuscript to revise grammatical errors and sentence structure throughout. Please ensure appropriate detail provided in methods and results as outlined by reviewer comments and review and revise abstract accordingly. Please review tables 2 and 3 to ensure the data is clearly presented e.g. Table 2 states sex (female) = 18 (81.8) but it is not clear that this is n (%female).

Reviewers' comments:

Reviewer's Responses to Questions

**Comments to the Author**

1. Does this manuscript meet PLOS Digital Health’s publication criteria? Is the manuscript technically sound, and do the data support the conclusions? The manuscript must describe methodologically and ethically rigorous research with conclusions that are appropriately drawn based on the data presented.

Reviewer #1: Yes

Reviewer #2: Yes

Reviewer #3: Yes

2. Has the statistical analysis been performed appropriately and rigorously?

Reviewer #1: Yes

Reviewer #2: Yes

Reviewer #3: Yes

3. Have the authors made all data underlying the findings in their manuscript fully available (please refer to the Data Availability Statement at the start of the manuscript PDF file)?

Reviewer #1: Yes

Reviewer #2: Yes

Reviewer #3: No

4. Is the manuscript presented in an intelligible fashion and written in standard English?

Reviewer #1: Yes

Reviewer #2: Yes

Reviewer #3: Yes

5. Review Comments to the Author

Reviewer #1: Despite the feasibility nature of the present study, the paper of Benavent D et al. is the description of one of the most interesting real-life experiences of the use of digital health in the field of rheumatology in Europe. 

It represents also a good example of application of telemedicine in the south of Europe, which is not obvious. 

The data on a longer period of time will be interesting in order to understand the real benefit on the control of the diseases, such as the application of this technology in other diseases.

I think that this paper is suitable of publication after a minimal revision.

Despite the fact that the number of the final enrolled patients is small, the results are interesting and supports the use of mHealth for the cure of chronic arthritis.

Considering that the study is a good example of the application of remote care in clinical practice, I suggest to add somewhere a comment on the EULAR Ptc on telemedicine (http://dx.doi.org/10.1136/annrheumdis-2022-222341).

-Patients

I would better clarify the clinical manifestations of enrolled patients with SpA which is a more heterogeneous disease than RA.

Spondylitis or peripheral arthritis?

Any extra-articular manifestations (also for RA)?

I have noticed that patients enrolled in the study had a milder disease in comparison with that of SoC group. Explain the reason which is not obvious.

No patients in b/tsDMARDs? Why? Was it a specific choice?

-Physicians

A curiosity: the physicians who were available for the remote consultations were the same who followed up patients with f2f visits? Or there was a dedicated personnel for the digital part?

-I would do a comment about the % of participants which was low if we think about the total cohort that researchers had in mind. Do you think that some cultural barrier limited the participation?

I think that in this type of study, a comment on the socio-cultural environment has to be done in order to better understand the data. A comparison with other similar experiences but in other Countries can be interesting.

-App

Is it thinkable to use this app in a large scale?

A comment on the costs of this feasibility study can be appreciated.

Was this app used in the past for other health applications? If yes, spend few words on this.

-Figure 2. I am not sere it is useful... it is not clear enough.

Reviewer #2: PLOS review

Monitoring Chronic Inflammatory Musculoskeletal Diseases Mixing Virtual and Face-to-Face Assessments - Results of the Digireuma Study

This study describes the development and pilot study of a Mixed Attention Model for remotely monitoring rheumatoid arthritis and spondyloarthritis. This is an important topic due to the large number of people affected and the intensive follow-up necessary to best handle these conditions.

Abstract:

The word “anytime” in the second to last sentence in the “Methods” subsection should be “any time.” In the “Results” subsection, please report the NPS rather than an overall rating with stars. The NPS is a standardized measurement tool that can be compared with other studies. There is nothing in the methods section that describes what question was asked of participants that would result in a star rating. In the “Conclusion” subsection, consider changing the first sentence to read “The use of the digital health solution is feasible in clinical practice to monitor ePROs for RA and SpA.”

Background:

Rather than saying “thousands of people worldwide” please provide a more accurate estimated number. I would guess we are talking about hundreds of thousands. In the final sentence of the first paragraph, do you mean “mHealth” rather than “m-pHealth”? It is not clear what the latter term means. In the final sentence of this section, please be consistent with the statistical analysis paragraph, which mentions a “feasibility” study, rather than a “proof-of-concept” study.

Methods:

Please consider reorganizing this section. The development of the protocol and the application seem to be mixed with the pilot test and the evaluation of the pilot test. I would suggest the following:

1. Development of protocol and application (focus group, Mixed Attention Model). It wasn’t clear whether the semi-structured interviews were part of the development process, or were done after the feasibility study.

2. Pilot test of the application (Digireuma Prospective Study)

3. Evaluation of the pilot test (Statistical analysis)

The heading “Identification of unmet needs…” should probably be in italics.

The methodology for the focus group needs to be better explained. How were people selected for the focus group? Who developed the focus group guide? Who facilitated the group? Was the discussion in the focus group transcribed and analyzed for themes?

When were the semi-structured interviews completed? Were they done before or after the prospective study? The paper mentions the results of semi-structured interviews that were completed with participants following the study. Are these the same ones? If there are two different instances of semi-structured interviews, please describe how recruitment was done for the interviews.

Please provide examples of the motivational messages mentioned in the “MAM Implementation-Adaptation…” section. How was the input from patients obtained for the agreed-on set of ePROs?

In the section titled “Digireuma Prospective Study” please add the timeframe for the recruitment of patients. In the paragraph on Follow-up, please indicate how long patients were asked to use Adhera. It is not until the discussion section that I saw that there was a three-month period of use. In the final sentence of this section, you mention semi-structured telephone interviews, but not when they were completed and by whom. Were the interviews transcribed? Were they analyzed for themes?

Results:

This section suffers from the same confusion between the focus group and the qualitative interviews as was in the methods section. The results section on the focus group refers to Table 1, which is titled to specify that these features were requested in qualitative interviews. Focus groups and qualitative interviews are not the same thing, and the paper should be consistent in referring to them.

At the end of the section titled “Pilot-Testing Study,” you mentioned that patients in the Digireuma group were younger than those in the general cohort twice.

The NPS is computed by subtracting the percentage of detractors from the percentage of promoters, 64% -7% = NPS of 57. It is not generally reported as fractions. What item was used for the general star rating of the app? This is never described in the methods section.

Discussion:

Why was the average age so much lower in the pilot test group than in the focus group?

At the end of the first paragraph in this section, you mention 4 out of 5 patients completing data using the digital solution, but Table 3 indicates a range of 69 to 88.9% for completion of ePROs. You state that “most” of the interventions were remote. Please provide a percentage or the actual number. You state that the majority of patients were considered promoters, but you should mention that only 14 patients rated the app using the NPS.

Please type out the source of the acronym “EULAR PtC.”

In the second to last paragraph, you mention NPW. Did you mean NPS?

Reviewer #3: Benavent et al. reported the performance of a hybrid (virtual and face-to-face) monitoring strategy for personalized care in rheumatoid arthritis (RA) and spondyloarthritis (SpA), which has implemented a mobile app to control the patient reported outcomes in pateints with SpA and RA. The paper is important given the general trend for the digitalization in healthcare and scarcity of data on digital solutions in rheumatology. Nevertheless, there are several points that have to be addressed by authors.

1. Authors claim „Thousands of people worldwide are affected by such RMDs as rheumatoid arthritis (RA) and spondyloarthritis (SpA)”. This is an underestimation, given the estimated global prevalence of RA of around 0.5% (Almutairi K, Nossent J, Preen D, Keen H, Inderjeeth C. The global prevalence of rheumatoid arthritis: a meta-analysis based on a systematic review. Rheumatol Int. 2021;41(5):863-877) and of SpA ranging between 0.2% to 1.6% (Stolwijk C, van Onna M, Boonen A, van Tubergen A. Global Prevalence of Spondyloarthritis: A Systematic Review and Meta-Regression Analysis. Arthritis Care Res (Hoboken). 2016;68(9):1320-1331). So, the phrase may give a reader a wrong impression, as there are hundreds of thousands to millions of people affected with the diseases, not thousands. 

2. Authors claim that 2 clinicians (DB, CP) were involved in the focus group. I am struggling to identify who is referred to as CP, as there is no co-author listed with such initials.

3. Authors claim that 5 patients were involved in the focus group. Could the authors please clarify how the size of the focus group was determined to make sure it depicts the real-life picture, and how the participants were selected?

4. In the Results section, authors report that “The focus group included 5 patients with RA”. In the methods section, authors report that altogether 5 patients were included in the focus group. Does this mean that no patients with SpA were included in the focus group, so the SpA patients sis not participate in the selection of PROs and the frequency of their assessment for SpA (unlike RA)? If so, could the “Identification of unmet needs- Focus group” section be corrected accordingly?

5. In the Sub-Chapter “Conceptualization of a Mixed Attention Model”, authors provide the reference as ISBN:978-84-09-38502-7. Unfortunately, I am struggling to find the source they are referring to both using search engines and ISBN identificators. Could author please cite the book they are referring to in a more convenient way or provide the description of face-to-face visits and virtual follow-ups in detail in the text? 

6. From the Supplementary Table 1, it is clear that different patient-reported outcomes were not accessed at once but on different days. Could the authors please provide the reason for such assessment and mention the problems related to it in the discussion (e.g. problems with potential use of data to calculate disease activity indices such as ASDAS, as it supposes that PROs included in the index – BASDAI and PtGA – are registered at the same moment of time)?

7. In the Methods section, authors write that Bath Ankylosing Spondylitis Functional Index (BASFI) was administered to1 the patients. At the same time, BASFI is not included in the Supplementary Table 1. Could the authors please address this discrepancy?

8. Authors claim: “The characteristics of the patients in the Digireuma study were compared to the baseline characteristics of a sample cohort from the UCT, matched by disease.” Could the authors please specify what cohort are they referring to and characterize it, as the abbreviation “UCT” is not explained in the text and no reference provided?

9. In the result section, authors report that the mean age of the included patients (48± 12 and 42± 9 years in the RA and SpA groups) was significantly different from the mean age of the focus group (62 ± 16.7 years). Could the authors please comment on such a discrepancy and whether they see it as an important one?

10. Authors report that there was a significant difference in the number of interactions between RA and SpA groups, especially in the number of completed questionnaires (RA - 2,178 questionnaire items, SpA - 1637 questionnaire items; patients with RA completed a median of 9.5 (4.3, 15.8) ePROs during the 3-months, while patients with SpA completed a median of 3 ePROs (1, 12)), although the frequency of the administered questionnaires was the same between the groups (every second week), and SpA patients had more PROs to fill in (so more answers expected). Could the authors please explain such a difference between the patient groups?

11. As authors explore feasibility of the proposed digital assessment, I would be very much interested in the “retention rate”, which authors unfortunately report only partially (of 46 patients, 5 did not download the application, and 4 did not submit at least one entry). Could the authors please provide the number of patients who completed all the prescribed PROs, as well as the number of patients who have completed at least 75% and at least 50% of the prescribed PROs?

12. For the feasibility assessment, physician’s perspective is missing. Could authors please comment on the working load of “the digital clinician”: how time consuming for was it to check the data provided by patients and to contact patients?

13. As can be seen in the Table 2, both patients with RA and SpA included in the study were having a significantly milder disease as compared to the standard care (e.g. DAS28 2.9 vs. 5.2 for RA patients, BASDAI 2.1 vs. 6 for SpA patients). Therefore, the possibility to generalize the findings to the general population of patients with RA and SpA, as well as the feasibility of the digital assessments (as more severe patients would report more flares) are questionable. Could authors please comment on that issue?

14. Although PROs are used widely to determine the disease activity of both SpA and RA, using PROs only for defining disease activity (and therefore treatment strategy) may have its own disadvantages. It is well known that most common manifestations of both RA and SpA (such as pain) may have other reasons apart from SpA and RA itself, such as concomitant fibromyalgia, osteoarthritis, depression, other conditions. A significant discrepancy between self-reported and physician-measured arthritis, especially for swollen joint count, was also previously demonstrated (Cheung PP, Gossec L, Mak A, March L. Reliability of joint count assessment in rheumatoid arthritis: a systematic literature review. Semin Arthritis Rheum. 2014;43(6):721-729). How did authors address this issue? Were patients instructed to differentiate between the manifestations of the rheumatic disease and other causes of musculoskeletal symptoms? Were comparisons performed between patient-reported and physician-detected joint problems?

6. PLOS authors have the option to publish the peer review history of their article (what does this mean?). If published, this will include your full peer review and any attached files.

**Do you want your identity to be public for this peer review?** For information about this choice, including consent withdrawal, please see our Privacy Policy.

Reviewer #1: Yes: SILVIA PIANTONI

Reviewer #2: No

Reviewer #3: Yes: Mikhail Protopopov

---

## [Decision Letter · Decision Letter 1]

9 Nov 2022

Monitoring Chronic Inflammatory Musculoskeletal Diseases Mixing Virtual and Face-to-Face Assessments - Results of the Digireuma Study

PDIG-D-22-00071R1

Dear Dr Benavent,

We are pleased to inform you that your manuscript 'Monitoring Chronic Inflammatory Musculoskeletal Diseases Mixing Virtual and Face-to-Face Assessments - Results of the Digireuma Study' has been provisionally accepted for publication in PLOS Digital Health.

Best regards,

Belinda Lange

Academic Editor

PLOS Digital Health

Thank you for providing a comprehensive and considered response to the reviewer comments. In preparation for review of final version of proof for publication:

1) In Results section (page 30) under 'focus group' subheading, in the first line, please spell out the numbers five and two

2) Please review and reconsider if Table 1 should be labelled as a figure rather than a table

Reviewer Comments (if any, and for reference):

Reviewer's Responses to Questions

**Comments to the Author**

1. If the authors have adequately addressed your comments raised in a previous round of review and you feel that this manuscript is now acceptable for publication, you may indicate that here to bypass the “Comments to the Author” section, enter your conflict of interest statement in the “Confidential to Editor” section, and submit your "Accept" recommendation.

Reviewer #1: All comments have been addressed

Reviewer #2: All comments have been addressed

Reviewer #3: All comments have been addressed

2. Does this manuscript meet PLOS Digital Health’s publication criteria? Is the manuscript technically sound, and do the data support the conclusions? The manuscript must describe methodologically and ethically rigorous research with conclusions that are appropriately drawn based on the data presented.

Reviewer #1: Yes

Reviewer #2: Yes

Reviewer #3: Yes

3. Has the statistical analysis been performed appropriately and rigorously?

Reviewer #1: Yes

Reviewer #2: Yes

Reviewer #3: Yes

4. Have the authors made all data underlying the findings in their manuscript fully available (please refer to the Data Availability Statement at the start of the manuscript PDF file)?

Reviewer #1: Yes

Reviewer #2: Yes

Reviewer #3: Yes

5. Is the manuscript presented in an intelligible fashion and written in standard English?

Reviewer #1: Yes

Reviewer #2: Yes

Reviewer #3: Yes

6. Review Comments to the Author

Reviewer #1: I think that the Authors have addressed all my comments.

Reviewer #2: No additional comments.

Reviewer #3: (No Response)

7. PLOS authors have the option to publish the peer review history of their article (what does this mean?). If published, this will include your full peer review and any attached files.

**Do you want your identity to be public for this peer review?** For information about this choice, including consent withdrawal, please see our Privacy Policy.

Reviewer #1: **Yes: **Silvia Piantoni

Reviewer #2: No

Reviewer #3: **Yes: **Mikhail Protopopov
